# Proton pumping accompanies calcification in foraminifera

Takashi Toyofuku[1,*], Miki Y. Matsuo[2,*], Lennart Jan de Nooijer[3,*], Yukiko Nagai[1], Sachiko Kawada[1], Kazuhiko Fujita[4], Gert-Jan Reichart[3,5], Hidetaka Nomaki[6], Masashi Tsuchiya[1], Hide Sakaguchi[2] & Hiroshi Kitazato[7]

Ongoing ocean acidification is widely reported to reduce the ability of calcifying marine organisms to produce their shells and skeletons. Whereas increased dissolution due to acidification is a largely inorganic process, strong organismal control over biomineralization influences calcification and hence complicates predicting the response of marine calcifyers. Here we show that calcification is driven by rapid transformation of bicarbonate into carbonate inside the cytoplasm, achieved by active outward proton pumping. Moreover, this proton flux is maintained over a wide range of $pCO_2$ levels. We furthermore show that a V-type $H^+$ ATPase is responsible for the proton flux and thereby calcification. External transformation of bicarbonate into $CO_2$ due to the proton pumping implies that biomineralization does not rely on availability of carbonate ions, but total dissolved $CO_2$ may not reduce calcification, thereby potentially maintaining the current global marine carbonate production.

[1] Department of Marine Biodiversity Research (B-DIVE), Japan Agency for Marine-Earth Science and Technology (JAMSTEC), Natsushima-cho 2-15, Yokosuka 237-0061, Japan. [2] Department of Mathematical Science and Advanced Technology (MAT), Yokohama Institute for Earth Sciences (YES), Japan Agency for Marine-Earth Science and Technology (JAMSTEC), 3173-25, Showa-machi, Kanazawa-ku, Yokohama-City, Kanagawa 236-0001, Japan. [3] Department of Ocean Systems, NIOZ-Royal Netherlands Institute for Sea Research and Utrecht University, Landsdiep 4, 1797 SZ 't Horntje, The Netherlands. [4] Department of Physics and Earth Sciences, Faculty of Science and Tropical Biosphere Research Center, University of the Ryukyus, 1 Senbaru, Nishihara, Okinawa 903-0213, Japan. [5] Department of Earth Sciences – Geochemistry, Faculty of Geosciences, Utrecht University, P.O. Box 80.021, 3508 TA Utrecht, The Netherlands. [6] Department of Biogeochemistry, Japan Agency for Marine-Earth Science and Technology (JAMSTEC), Natsushima-cho 2-15, Yokosuka 237-0061, Japan. [7] Tokyo University of Marine Science and Technology, Konan 4-5-7, Minato-ku, Tokyo 108-8477, Japan. * These authors contributed equally to this work. Correspondence and requests for materials should be addressed to T.T. (email: toyofuku@jamstec.go.jp).

Marine calcification plays an important role in the global carbon cycle and it is estimated that up to 90% of all carbon currently buried in the seafloor results from biogenic calcium carbonate production[1,2]. On geological timescales, $CaCO_3$ production and $pCO_2$ are largely decoupled as alkalinity is added to the ocean from weathering. However, on time scales up to hundreds of years, calcification increases $pCO_2$ as the lowered alkalinity shifts the inorganic carbon speciation towards $CO_2$. Results from culturing experiments mimicking ocean acidification showed contrasting responses of calcification: calcification was reduced in some species, whereas others were not affected[3]. A large portion of open ocean calcium carbonate production, between 20 and 50%, derives from perforate foraminifera[4,5]. Despite its clear importance for the global carbon cycle, the physiological processes responsible for calcification in foraminifera are poorly understood. The key to understanding foraminiferal calcification centres on the relation between carbon speciation in seawater and preferential uptake of these chemical species ($CO_2$, bicarbonate and/or carbonate ions)[3–6].

Foraminifera build their tests by sequentially adding chambers. When foraminifera add a new chamber, the protoplasm does not contain sufficient building blocks/materials for calcifying an entirely new chamber. Limited availability of carbonate ions in seawater dictates that foraminifera may require unrealistic volumes of seawater to produce new calcium carbonate[6]. Hence, calcification by foraminifera has been hypothesized to involve intracellular storage of calcium ions and inorganic carbon[7], pH manipulation[6,8,9] and active calcium[10] and/or magnesium pumping[11]. These results and the variety of postulated mechanisms for foraminiferal calcification[6,7,10–13] make it challenging to reliably predict response to changes in marine inorganic carbon perturbations. Carbon and calcium uptake mechanisms and rates have been based on a combination of (scanning and transmission electronic) microscope observations[14–16], isotope labelling[17], microelectrode measurement[9,18] and analysis of the elemental and stable isotopic composition of foraminiferal calcite[11,12]. Recently, this has been complemented by applying fluorescent indicators visualizing ion fluxes inside actively calcifying specimens[6,8,19–21]. Imaging extracellular pH around benthic perforate foraminifera allows carbon speciation during calcification outside these foraminifera to be assessed. Although microelectrode analyses previously shows potential changes in carbon speciation outside the cytoplasm[9], it remains to be quantified whether, and to what degree, different carbonate species contribute to calcification.

Here we show external pH change throughout the calcification of perforate foraminifera Ammonia sp., at a range of $pCO_2$. Our results allow the calculation of proton fluxes and hence establish a quantitative calcification budget. Our physical model for calcification shows the dependence of foraminiferal biomineralization on the various inorganic carbon species present in seawater. We validate the importance of pH regulation on the foraminiferal calcification by application of a V-type $H^+$ ATPase inhibitor, which plays a key role in aragonite production in scleractinian corals[22,23].

## Results

**External pH around foraminifera during chamber formation.** The first visualization of the extracellular spatial distribution of pH during chamber formation shows a strong decrease in external pH surrounding specimens of the benthic non-symbiotic foraminifer Ammonia sp. (Fig. 1, Table 1 and Supplementary Movie 1). This decrease in pH is modest at the start of chamber formation and intensifies over time in all five specimens studied,

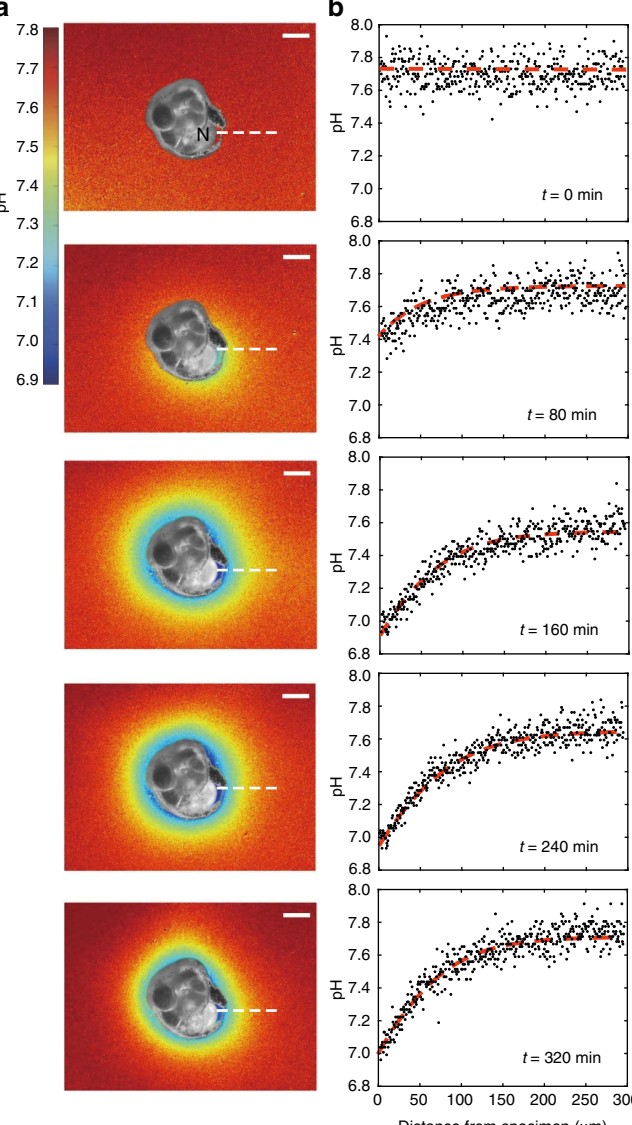

**Figure 1 | Reduction in pH during foraminiferal calcification.** Representative images showing the time-resolved decrease in pH of seawater surrounding a calcifying specimen of Ammonia sp. over a period of 320 min. The pH values are imaged using dissolved HPTS and reported on the seawater scale. The incubated specimen shows (**a**) the two-dimensional variability in pH around the shell when building a new chamber and (**b**) the translated, spatially integrated change in pH versus distance from the foraminifer along the white dotted line shown in **a**. At the start of calcification, surrounding pH is still ~7.8 outside the foraminifer, decreasing to 6.9 after 4 h and subsequently gradually increasing again 6 h after the onset of calcification. It is noteworthy that minimum pH values are found closest to the newly precipitated chamber (N). In addition, a zone of reduced pH encloses the complete shell, also where no new chamber is being produced. The gradient in pH, increasing with distance from the specimen, is mainly caused by protons diffusing away from the site where the new calcite is precipitated. Scale bars, 100 μm. The false-colour scale bar represents pH. The b/w foraminifer is superimposed on false-colour pH images.

decreasing to a minimum value of ~6.9 about 6 h after the start of calcification. The strongest pH decrease is observed closest to each organism and in particular near the newly forming chamber. Ultimately, after completion of the new chamber (on average

**Table 1 | Summary of pH imaging observations during chamber formation.**

| No. | Total time of chamber formation (h:mm) | The lowest pH during an event (h:mm) | Shell diameter (µm) | Calculated total proton flux (pmol) | Ambient pH (seawater scale) | Estimated $p\mathrm{CO_2}$ (µatm) |
|---|---|---|---|---|---|---|
| 1 | 4:30 | 7.1 (2:25) | 141 | 6 | 8.0 | 460 |
| 2 | 5:05 | 7.1 (3:25) | 216 | 17 | 8.0 | 460 |
| 3 | 6:05 | 7.0 (3:50) | 323 | 68 | 7.9 | 610 |
| 4 | 5:55 | 7.1 (2:55) | 166 | 6 | 7.8 | 790 |
| 5 | 4:45 | 6.4 (2:30) | 228 | 14 | 7.7 | 1,030 |
| 6 | 6:00 | 6.9 (2:52) | 260 | 58 | 7.6 | 1,320 |
| 7 | 4:55 | 6.4 (1:45) | 268 | 15 | 7.3 | 2,160 |
| 8 | 3:45 | 6.7 (2:45) | 243 | 4 | 7.3 | 2,760 |
| 9 | 4:05 | 6.7 (1:00) | 203 | 6 | 7.3 | 2,760 |
| 10 | 5:00 | 6.3 (1:15) | 186 | 5 | 6.8 | 9,010 |
| *With V type H$^+$ ATPase inhibitor* | | | | | | |
| 11 | 2:15 | 7.2 (1:20) | 231 | nd | 7.5 | 1,560 |
| 12 | 1:50 | 7.1 (1:05) | 256 | nd | 7.5 | 1,560 |
| 13 | 8:00 | 7.4 (8:00) | 308 | nd | 7.5 | 1,560 |

Reproducibility of pH value < 0.15 and total alkalinity of the solution is $2.330 \pm 15\,\mu\mathrm{mol\,kg^{-1}}$.

between 6 and 12 h after calcification commences), external pH returned to ambient, pre-chamber formation values.

This decrease in external pH was observed over a wide range of $p\mathrm{CO_2}$ (Table 1) and the reduction in pH compared with that of the ambient seawater was relatively constant over the experimental conditions. With a reduction in seawater pH by ∼1 unit, the pH in the foraminiferal microenvironment also decreased by ∼1 unit (Table 1). There was no clear relation between the foraminiferal size and the pH reduction, although specimens with the largest diameter were associated with the highest total proton flux (Table 1).

After addition of the V-type H$^+$ ATPase inhibitor Bafilomycin A$_1$ at the onset of chamber formation, no clear external pH gradient develops, indicating a negligible proton flux. Occasionally, a very small decrease in external pH was observed during incubation with Bafilomycin (Table 1 and Supplementary Fig. 1). During these incubations, foraminifera produced very thin chamber walls, consisting mainly of the organic sheet produced at the beginning of new chamber formation (Supplementary Fig. 2).

## Discussion

Combining time-resolved external pH recordings with two-dimensional pH gradient observations (lowest proximal to the newly formed chamber at 160 min; Fig. 1), allows calculating total proton flux ($Q_{\mathrm{H}}$) from the site of calcification (SOC) to the specimen's microenvironment (Fig. 2). The cumulative proton flux increase is relatively linear over time and results in a final cumulative proton flux. We found that the observed radial decrease in $[\mathrm{H^+}]$ is well approximated by the second type of the modified spherical Bessel function, implying that the protons diffuse away from the foraminifer and that a proportion of them is consumed by carbonation during diffusion (for example, by the reaction with $\mathrm{HCO_3^-}$ to form $\mathrm{CO_2}$ and $\mathrm{H_2O}$). Proton flux originating from within the foraminifer is calculated by fitting the Bessel function and using Fick's law (see 'Modelling proton flux' in Methods). The shape of the foraminifer is here considered spherical with a radius $R = 100\,\mu\mathrm{m}$ and the proton flux is regarded homogenous over the complete specimen's surface. Total proton flux thus integrates flux over the surface of the protective envelope (estimated to be $0.03\,\mathrm{mm^2}$; Fig. 1). For an average decrease in pH (0.5 at the surface of the specimen and 0.1 at a distance of $100\,\mu\mathrm{m}$), an indicated specimen releases protons by an average flux $Q_{\mathrm{H}} = 0.014\,\mathrm{nmol\,h^{-1}}$. The final cumulative proton flux (4–68 pmol; Table 1) is in the same order

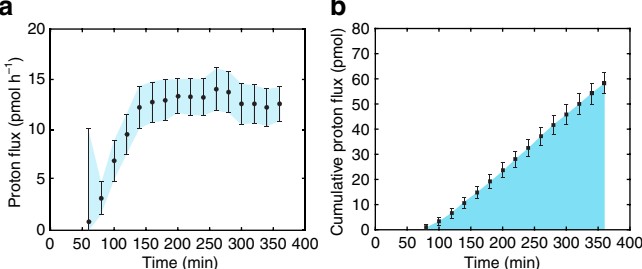

**Figure 2 | Calculated proton flux from the reduction in environmental pH.** (**a**) Time series of proton flux during chamber formation and (**b**) the corresponding cumulative proton flux (Specimen no. 3 in Table 1). These estimates are based on analysis of the pH image series by theoretical fitting of the decreased pH as a function of distance from the foraminifer. Error bars indicate s.d.

of magnitude as the total dissolved inorganic carbon (DIC) flux and 0.5 of the total amount of $\mathrm{Ca^{2+}}$ (2–34 pmol) necessary for the calcification of a new chamber. For a hemispherical chamber with a diameter of 20–50 µm, a wall thickness of 3 µm and a porosity of 25%, the required $\mathrm{Ca^{2+}}$ equals ∼30–210 pmol[13]. The similarity in fluxes may indicate that these processes are directly coupled, but may also be coincidental.

The observed decrease in pH outside the individual's shell during calcification of benthic foraminifer *Ammonia* sp. implies that this foraminifera actively pump protons out of their protoplasm, with the flux independent of initial external pH (Fig. 3). Observation in the presence of the inhibitor Bafilomycin A$_1$ suggests that a V-type H$^+$ ATPase is responsible for the proton transport (Supplementary Fig. 1). This is in line with earlier pH observations inside[21] and outside[9] calcifying foraminifera. The impact of decreased pH outside the foraminifer shifts inorganic carbon speciation as $\mathrm{CO_3^{2-}}$ is transformed into $\mathrm{HCO_3^-}$ and bicarbonate into $\mathrm{CO_2}$ (Fig. 2). Within the SOC, elevated pH[8,21] results in the opposite shift in speciation as $\mathrm{HCO_3^-}$ and $\mathrm{CO_2}$ are transformed into $\mathrm{CO_3^{2-}}$ (Fig. 3). Hence, calcification is characterized by strong gradients in pH and $p\mathrm{CO_2}$ between the SOC and the foraminiferal microenvironment (from 6.9 to 9 for pH and ∼7,200 µatm to < 20 µatm for $p\mathrm{CO_2}$). Involvement of respired $\mathrm{CO_2}$ may be responsible for part of the lowered pH. However, such a process is unlikely affected by the presence of Bafilomycin A$_1$, which prevented a clear pH decrease

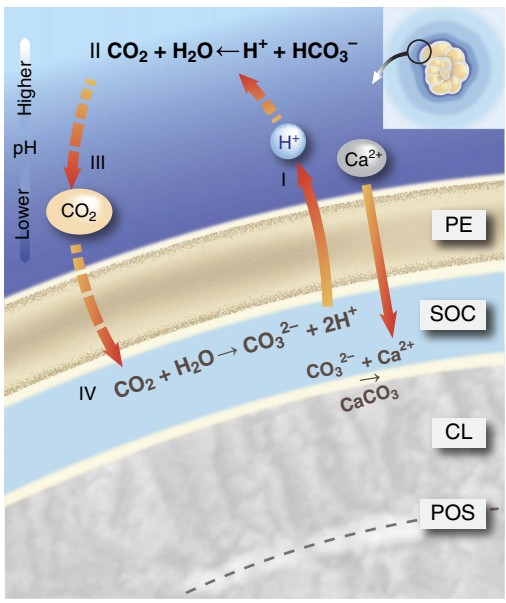

**Figure 3 | Proton pumping-based model of foraminiferal calcification.**
During calcification of a new calcitic layer (CL) on a primary organic sheet (POS), the protective envelope (PE) separates the growing calcite surface from the surrounding seawater. The chemical composition at the SOC, created by the PE, is characterized by active, outward proton pumping (I). The reduced pH in the foraminiferal microenvironment shifts the inorganic carbon speciation (II), thereby increasing $pCO_2$ directly outside the PE. The large gradient in $pCO_2$ across the PE results in diffusion of $CO_2$ into the SOC (III). Once inside, the $CO_2$ reacts to form $CO_3^{2-}$ due to the high pH (IV) sustaining $CaCO_3$ precipitation by reacting with the $Ca^{2+}$. The reduction in pH is seen over the entire foraminifer (inset), suggesting that this model applies to the complete surface of the shell of a rotalid foraminifer producing a new chamber.

during chamber formation. We therefore infer that the impact of respired $CO_2$ on the lowered external pH is minor.

As $CO_2$ diffuses easily across cell membranes compared to $HCO_3^-$, the large $pCO_2$ gradient results in a flux of carbon dioxide into the foraminifer (Fig. 3). The high pH at the SOC locally increases saturation state and hence promotes calcification (Fig. 3). Inside the specimen, excess protons from the conversion of $CO_2$ into (bi)carbonate help sustain $CaCO_3$ production by reacting with the $Ca^{2+}$ transported inwards[10] and the continued proton flux outside of the foraminifer (Fig. 3).

Modelling proton pumping to mimic the observed pH gradient outside the specimen over time (Fig. 1) implies that more than half of the protons are consumed by the reaction with bicarbonate. Therefore, the calculated increase in $pCO_2$ converts between 25 and 50% of all DIC into carbon dioxide directly outside the foraminifer. The exact value converted depends on the appropriate dissociation constant for the conversion between $CO_2$ and $HCO_3^-$, and on the exact pH of the foraminifer's microenvironment. The rate at which this $CO_2$ is taken up by the foraminifer depends on the thickness of the pseudopodial envelope across which the $CO_2$ diffuses and the constant rates for the reactions of the inorganic carbon species at the SOC (Fig. 3). The hydration of $CO_2$ to form bicarbonate and a proton is relatively slow and could therefore limit calcification rates. The slow kinetics of this reaction may however be 'bypassed' by $CO_2$ reacting with $OH^-$ at the SOC[24]. Alternatively, the conversion rate may be increased by the presence of specialized enzymes like carbonic anhydrase, which are known/suggested to be involved in the calcification of other

marine calcifyers including corals[23], coccolithophores[25] and bivalves[26]. Although not relevant for the fluxes calculated here, ultimately a more precise characterization of the chemical composition at the SOC is necessary to show the relative contribution of these pathways to the overall conversion of $CO_2$ into carbonate.

Culture studies using planktonic foraminifera show that the carbon isotopic composition ($\delta^{13}C$) of newly formed chambers decreases with increasing $CO_3^{2-}$ or increasing pH[27]. In equilibrium, the $\delta^{13}C$ of dissolved bicarbonate is enriched with respect to that of the total dissolved inorganic carbon pool[28]. Assuming that foraminifera precipitate their shell in equilibrium with DIC in seawater, the generally negative foraminiferal shell's $\delta^{13}C$ values[29,30] suggest that bicarbonate is not a direct carbon source for calcification. Carbon dioxide is the only inorganic carbon species isotopically depleted compared with the total inorganic carbon pool. This is in line with the here suggested carbon uptake via $CO_2$ at lower pH than that of the culture medium. This role of $CO_2$ on controlling carbon isotopic values in foraminiferal carbonate is similar to that proposed by previous studies, except that we here propose that the source of this $CO_2$ is through direct pH manipulation rather than via respiration and/or $CaCO_3$ precipitation[27,31].

Our results suggest that calcification is not directly coupled to the presence of carbonate ions and hence does not depend on the calcite saturation state[32,33]. Instead, foraminiferal calcification would rely on total inorganic carbon concentration. This uncoupling of saturation state and calcification explains the moderate response of many foraminiferal calcification rates to experimentally induced ocean acidification[34] and the occurrence of diverse foraminiferal communities at conditions that are undersaturated with respect to calcite, but have high DIC concentrations[35]. Under such conditions, foraminifera are able to acquire sufficient carbonate ions through proton pumping and inward $CO_2$ diffusion to sustain chamber addition. The foreseen reduction in pH (from 8.1 today to ~7.8 at the end of the twenty-first century[36]) by increased oceanic $CO_2$ uptake is relatively small compared with the pH decrease in the foraminiferal microenvironment (down to 6.9 in Fig. 1) during calcification. The decrease in ambient pH (Table 1) does not noticeably affect the strong decrease in pH in the foraminiferal microenvironment as a result of calcification. Hence, a relatively moderate decrease in pH may not impair foraminiferal calcification, especially as DIC increases at the same time. Ocean acidification may still affect calcification indirectly (for example, through altered metabolism). These effects probably differ considerably between species, which may explain the observed large interspecific variation in foraminiferal response to reduced pH[34]. Proton pumping is found to be the fundamental strategy by which a variety of organisms produce calcium carbonate[37]. The high internal pH[8,21] and large internal–external pH difference associated with foraminiferal calcification (Table 1, Fig. 1) predicts that they may well produce more carbonate ions at the SOC under elevated $pCO_2$ (ref. 37). The partial decoupling between seawater pH and calcification shown here implies a reduced buffering capacity of the ocean with ongoing increases in atmospheric $CO_2$ concentrations, as calcification of this species does not necessarily decrease with ongoing acidification.

*Ammonia* is an infaunal genus widely applied as a bioindicator in neritic environments[38]. Despite limited knowledge regarding pH variability in pore waters, laboratory observations have shown that pH is variable around 0.6 units within the uppermost 2 mm of the sediment, deeper layers (>4 mm) experience smaller pH fluctuations (<0.2) on the timescale of hours[39]. Our results suggest that calcification of this species is unlikely affected

by such variations in ambient pH, as the foraminifer-induced pH changes exceed those occurring naturally.

## Methods

**Specimens.** Culture experiments and microscope observations were performed at the Japan Agency for Marine-Earth and Technology (JAMSTEC) laboratory, Yokosuka, Japan. The living specimens were collected from brackish water salt marsh sediments of Hiragata-bay, Natsushima-cho Yokosuka (35.3226°N, 139.6347°E). *Ammonia* sp. was used for the experiments, a benthic, hyaline, cosmopolitan species. Living specimens were isolated and cleaned from excess sediment and debris, transferred to filtered (0.2 μm) seawater and placed in a Petri dish. The dishes were maintained at 20 °C in filtered seawater with a pH of ∼7.9 and a $p$CO$_2$ of ∼550 μatm. Once a week, the seawater was replaced and living micro algae (*Dunaliella tertiolecta*) were added as food.

Ambient pH distributions were visualized around foraminiferal specimens that were starting to form a new chamber. We identified specimens close to forming a new chamber by the presence of excess fluffy material (for example, clastics and algae), forming a protective cyst, surrounding a fan-like pseudopodial network in the shape of a new chamber. At that moment, an organic membrane is expanding on the pseudopodial network, delineating the shape of the soon-to-be-built chamber. This organic membrane, also known as the primary organic sheet, serves as a template on which the first calcite of the new chamber precipitates. Specimens are cultured within 35 mm glass base dishes (3910-035, Iwaki glass).

**Observation settings.** For ambient pH imaging, pH indicator HPTS (pyranine 8-Hydroxypyrene-1,3,6-trisulfonic acid trisodium salt, H1529, Sigma-Aldrich) was dissolved to a final concentration of 20 μM[20]. This concentration of HPTS is known to be harmless to foraminiferal behaviour and does not noticeably impair their calcification process[20]. Total alkalinity of the solution is determined by pH method[40,41]. The observations were carried out with ten individuals under various pH/$p$CO$_2$ conditions. Individuals were incubated in the solution for 10 min before starting observations under room temperature (∼23 °C). The individuals were then observed under an inverted fluorescent microscope (Zeiss Axio Observer Z1, Germany).

Three individuals were additionally incubated with Bafilomycin A$_1$, a V-type H$^+$ ATPase inhibitor (BVT-0252, BioViotica). These incubations were done to investigate the influence of H$^+$ ATPases on calcification (see similar approach in scleractinian corals[22]). Bafilomycin A$_1$ was dissolved to a final concentration of 1 μM in seawater with 20 μM HTPS[42]. The specimens were placed in the solution only during chamber formation. All three specimens were observed trying to form a new chamber in the presence of Bafilomycin A$_1$.

**Optical settings.** Fluorescent filter cubes were used to detect pH signals from HPTS ($\lambda^{410}$exc = 395–425 nm, $\lambda^{470}$exc = 460–480 nm, $\lambda$em = 510–560 nm). Time-lapse images were captured every 5 min by a digital camera attached to the microscope using a standard software package (Axiovision, Version 4.6). Grey scale images representing different emission wave length intensities were exported as TIF files. Subsequently, ratiometric pH images were calculated by dividing $\lambda^{470}$em by $\lambda^{410}$em for each pixel, using a custom calibration curve[21] (Supplementary Fig. 3). The $p$CO$_2$ of each medium was estimated using the CO2SYS software package[43] after determining pH of the media ratiometrically and using total alkalinity.

**Observation management.** The pH of HPTS solution is manipulated by CO$_2$ bubbling just before the experimental incubation. The pH of the solution was continuously monitored by a pH meter (Thermo Scientific Orion 5-star Plus) equipped with a glass electrode (Thermo Scientific, PrpHecT ROSS Micro Combination pH electrode 8220BNWP), to ensure the appropriate amount of CO$_2$ was added. The pH values are indicated with the seawater scales.

The natural medium was replaced by seawater containing HPTS solution three times by removal of the seawater with a Pasteur pipette and subsequent addition of the HPTS-containing seawater. The pipetting was done very gently to avoid disturbance of any foraminiferal activities or to minimize gas exchange. The water's surface was covered by a cover glass to prevent gas exchange between water and air during observation. The pH was increased until the equivalent state reached the laboratory's atmospheric $p$CO$_2$ if the cover had not been used.

**Modelling proton flux.** First, we considered a model of proton release for a foraminifer. For simplicity, we assume that the foraminifer is spherical with radius $R$ and it is covered by a thin protective envelope. It is assumed that protons are released from the protective envelope and outside the foraminiferal cell and protons diffuse, and at the same time are consumed due to the carbonation reaction [H$^+$] + [HCO$_3^-$] → [H$_2$CO$_3$]. The reverse reaction is assumed not to occur, which is realistic due to the relatively low pH outside the specimen. With these assumptions, the proton concentration outside the foraminiferal cell can be

calculated using a diffusion equation with added consumption:

$$\frac{\partial}{\partial t}[\text{H}^+] = D_\text{H}\nabla^2[\text{H}^+] - \mu([\text{H}^+] - [\text{H}^+]_\infty), \qquad (1)$$

where $D_\text{H}$ is the diffusion constant of proton $D_\text{H} = 9.3 \times 10^{-5}\,\text{cm}^2\,\text{s}^{-1}$, $\nabla^2$ is the Laplacian operator and $\mu$ is the constant rate of the carbonation reaction. We solve this equation under the boundary conditions, $[\text{H}^+] = [\text{H}^+]_\infty$ at $r \to \infty$ and $[\text{H}^+] = [\text{H}^+]_\text{R}$ at $r = R$, where $[\text{H}^+]_\infty$ is the equilibrium concentration of protons in natural sea water and the value of $[\text{H}^+]_\text{R}$ is controlled by the foraminifer, depending on its developmental stage. When the foraminifer begins building a new chamber, $[\text{H}^+]_\text{R}$ becomes larger than $[\text{H}^+]_\infty$. After some time, equilibrium has been established and the spatial distribution of proton obeys the steady solution of equation (1) described by

$$[\text{H}^+] = \alpha\frac{K_{1/2}(kr)}{(kr)^{1/2}} + [\text{H}^+]_\infty, \qquad (2)$$

where $K_{1/2}$ is the second type of the modified Bessel function $K_a$ with $a = 1/2$ and $k \equiv \sqrt{\frac{\mu}{D_\text{H}}}$. The local radial flux of proton on the protective envelope is calculated using Fick's law,

$$J_\text{R} = -D_\text{H}\left(\frac{\partial[\text{H}^+]}{\partial r}\right)_{r=R}. \qquad (3)$$

When the shape of the foraminifer is spherically symmetric, the total flux is calculated by

$$Q_\text{H} = 4\pi R^2 J_\text{R}. \qquad (4)$$

Thus, the total flux $Q_\text{H}$ is determined by equations (2–4). We accordingly calculated the total proton flux of a foraminiferal specimen from its pH image. The nonlinear, least square fitting of the radial distribution of protons by equation (2) determines the values of coefficients $\alpha$, $k$ and $[\text{H}^+]_\infty$.

**Data availability.** The data in this study are available from the corresponding author on the reasonable request.

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

## Acknowledgements

We thank Mr Y. Tsuchiya (JAMSTEC) for video editing. We also thank Dr B. Mamo (The University of Hong Kong) for helpful discussions. This work was supported by JSPS KAKENHI Grant Numbers 22684027 (T.T.) and 25247085 (H.K.). This work was carried out under the programme of the Netherlands Earth System Science Center (NESSC).

## Author contributions

Scientific conception and experimental design: T.T., H.K. and H.S. Data acquisition and analysis: T.T., M.Y.M., L.J.d.N., Y.N. and S.K. Collected and processed data: T.T., M.Y.M. and L.J.d.N. Data interpretation: T.T., M.Y.M., L.J.d.N., K.F., G.-J.R., H.N., M.T. Wrote paper: T.T., M.Y.M., L.J.d.N. and G.-J.R. All authors discussed the results and commented on and revised the manuscript.

## Additional information

**Competing financial interests:** The authors declare no competing financial interests.

