## [Peer Review File · Nature Communications]

Reviewers' comments:

Reviewer #1 (Remarks to the Author):

This manuscript reports on pH variation in water surrounding a calcifying foraminifera through pH imaging during the calcification process. Based on those data, rates of proton flux were calculated and used to model ion flux / facilitation diffusion that are essential for the formation of calcite within the foraminifera cell. Importantly, the manuscript shows that decreases in carbonate saturation state (ω calcite) are unlikely to drive decreased calcification, as has generally been wrongly assumed in many ocean acidification studies, as calcification is driven by total DIC, and hence supported by the presence of bicarbonate and $p\text{CO}_2$. I believe that the ocean acidification community is increasingly understanding that calcification processes are not regulated by saturation state, but rather mediated by metabolic or other processes. However, decalcification or calcium carbonate dissolution, is likely a passive process that is related to ω .

I found the manuscript well written and important and I think that many researchers in the ocean acidification and calcification field will find the study to be useful.

The data as presented are strong, but I think this study would be stronger with direct demonstration of the molecular mechanisms that the authors surmise must be at work. For example, I would like to see some cellular imaging of proton pumps, such as the V-type H^+ ATPase that is known to be important in intra- and extracellular pH regulation (e.g., see work by Martin Tresguerres). Also, it would be important to use isotope (or other) labels to identify the source of carbon that is integrated into the newly formed calcite as proof of the modeled mechanism. Finally, I think that the inferences made in this study should be put into a natural ecological context for these organisms. For example, questions that I have include: How do natural rates of water flow around the foraminifera alter the extracellular pH gradient? What levels of natural variation in DIC do foraminifera experience, and how would this study predict variation in calcification as a result? For a paper published in a high profile journal such as Nature, I think expectations for the completeness of an analysis (data-model-validation) is not too high.

Reviewer #2 (Remarks to the Author):

Major Concerns:

1. I am unclear as to how many specimens the authors measured (line 76 mentions 5), but this should be clearly stated in the methods.
2. I'm not sure what this paper has to do with ocean acidification. The forams do decrease in pH at the site of chamber formation and calcification, but this study didn't test for incubation at various pH's. I suggest taking out references to this, except in the discussion sections.
3. Did the authors test for the effect the pH indicator had on calcification? If this has been done previously, please cite.
4. The implications of this study about calcification are modeled from inorganic carbonate chemistry measurements based on pH. I suggest the authors soften the language (Lines 183-185) to include the fact that these conclusions are from models, and not direct measurements of calcification.

Minor Concerns:

1. Please state what scale the indicator dye is measuring (NBS, seawater or total).
2. Please provide GPS points to where you collected your specimens, and in addition please provide information on culturing conditions before the observation (salinity, temperature, pH) if possible.

Reviewer #3 (Remarks to the Author):

Summary of key results

I'd like to start by congratulating the authors on conducting a set of simple but very elegant experiments using the pH sensitive fluorophore HPTS that visualize and quantify the proton efflux in the extracellular environment of foraminifers as they calcify new chambers. This is a novel set of results that makes a new and important contribution to understanding of biomineralization processes specifically in but not limited to foraminifers. It provides evidence for the likely use of 'massive' proton pumping to drive calcification, evidence to support a carbon concentration mechanism that could explain muted calcification responses to external environmental conditions, and also prima facie evidence for punctuated calcification with chamber addition in the studied specimens of *Ammonia* sp.

This is a highly original contribution and would be of general interest across the fields of climate change, biomineralization, cell biology, palaeoclimatology and palaeoceanography, among others.

Robustness and validity of Conclusions

Unfortunately the very real problem with the manuscript is the extent to which it overstates interpretations of findings as fact and asserts several unjustified conclusions about calcification responses to rising CO₂ levels. This is exemplified by the totally unsupported last sentence of the Abstract which states "These results suggest that increased CO₂ enhances calcification". Even if foraminifers did respond in this way, which most studies suggest they don't, the relatively small increase in surface ocean pCO₂ that would arise from a hypothetical increase in foraminifer calcification makes the following clause a gross overstatement of fact "thereby reducing the capacity of the ocean to take up atmosphere CO₂". To my knowledge virtually all studies of planktic and benthic perforate foraminifers show reduced (albeit slightly) calcification at higher pCO₂ and lower seawater pH and saturation state. Indeed this is acknowledged in the manuscript at line 187, where it is stated that many foraminifers exhibit moderate [negative] responses to experimentally induced ocean acidification. That the authors seem so keen to contradict and ignore this evidence in making their concluding statement is concerning.

In relation to the above comments the first three sentences of the Abstract summarize some of the well-known feedback response of the ocean-atmosphere carbon cycle. This is not the topic of the paper, but rather seems to reflect an effort to give the manuscript greater 'global' significance. This is disingenuous and indeed unnecessary in my view. The advance in understanding biomineralization and perhaps also for designing future experiments that could better address more fundamental organism responses to changing ocean chemistry is an important step forward.

Continuing in the same vein; I would suggest rewording of concluding paragraph beginning line 182. "Our results show that calcification is not directly coupled to the abundance of carbonate ions and saturation state of seawater." There then appears to be a limited basis for making the following statement, as there is no evidence supplied to suggest that calcification relies on the total inorganic carbon concentration, as much as this might or might not be true.

Possible additional experiments that address foraminifer responses to seawater DIC and saturation state?

The manuscript seems driven to generate a headline that states foraminifer calcification is not negatively affected by rising atmosphere pCO₂ and decreasing calcite saturation state, however there is little or no evidence generated here to justify these assertions nor to evaluate response to changes in foraminifer proton efflux responses to rising pCO₂ and decreasing saturation state. Based on the description of the experimental method it should be simple enough to undertake a small set of experiments to test how the visualized proton efflux (and calcification) changes with the DIC and pH of the seawater medium. This would

make for an more interesting study and would address some of the more fundamental responses that foraminifers exhibit to rising pCO₂ and decreasing saturation state.

Treatment of data and uncertainties, and interpretation of results

I wonder whether there is the possibility of a chicken-and-egg situation with the results. Specifically, could an equivalent proton flux be generated through calcification (driven by a mechanism other than proton pumping) combined with enhanced respired CO₂ production during chamber formation. How would the observed [H⁺] distribution around the foraminifer differ from this alternate scenario? Is a second observable in the seawater carbonate system needed to distinguish between these possibilities, or can the proton pump mechanism only explain the data?

What are the uncertainties of the proton flux and total flux QH estimates? Understanding the efficiency of converting CO₂ and HCO₃⁻ within the site of calcification (SOC) and thus the efficiency with which inorganic carbon is used for calcium carbonate production would be useful in the context of the proposed carbon concentration mechanism. How does the QH compare to the amount (in moles) of calcite produced during chamber addition? Specifically, does it account for the amount of calcite precipitated.

The model being proposed is that the proton efflux is used to raise the fCO₂ in the surrounding microenvironment to a level where it diffuses into the cell and SOC. While feasible, there is no direct evidence supporting this proposal. It would require external fCO₂ being > internal fCO₂ of the pseudopodial envelope in your model. This might be the case but the internal cellular fCO₂ would likely be quite high and so would create a partial barrier (slow point) that would not be particularly conducive to driving a large CO₂ flux from the high fCO₂ seawater to the low fCO₂ SOC. How accurate are the pH estimates obtained from the HPTS fluorophore calibration?

In relation to the model I note that there is also no evidence to substantiate the use of Ca²⁺, which otherwise implies a selective process for Ca uptake from external seawater in Figure 3 (as opposed seawater itself as a source of ions).

Overall the manuscript is well written and in a style that is easily understandable.

Additional minor comments

Chamber is typed incorrectly in Table 1

Line 132 .. should read .. Inside the site of calcification, elevated pH

K_a is noted to be the second type of the modified Bessel function but does not appear as such in (E.2).

Reply to Reviewers' comments:

Reviewer #1 (Remarks to the Author):

This manuscript reports on pH variation in water surrounding a calcifying foraminifera through pH imaging during the calcification process. Based on those data, rates of proton flux were calculated and used to model ion flux / facilitation diffusion that are essential for the formation of calcite within the foraminifera cell. Importantly, the manuscript shows that decreases in carbonate saturation state (ω calcite) are unlikely to drive decreased calcification, as has generally been wrongly assumed in many ocean acidification studies, as calcification is driven by total DIC, and hence supported by the presence of bicarbonate and $p\text{CO}_2$. I believe that the ocean acidification community is increasingly understanding that calcification processes are not regulated by saturation state, but rather mediated by metabolic or other processes. However, decalcification or calcium carbonate dissolution, is likely a passive process that is related to ω .

I found the manuscript well written and important and I think that many researchers in the ocean acidification and calcification field will find the study to be useful.

The data as presented are strong, but I think this study would be stronger with direct demonstration of the molecular mechanisms that the authors surmise must be at work. For example, I would like to see some cellular imaging of proton pumps, such as the V-type H^+ ATPase that is known to be important in intra- and extracellular pH regulation (e.g., see work by Martin Tresguerres).

We agree with the reviewer and have therefore added results from the incubation of foraminifera with the V-type proton-ATPase inhibitor Bafilomycin ($1 \mu\text{M}$; e.g. Bowman et al., 1988). This inhibitor prevents the transmembrane transport of protons and calcifying foraminifera are shown to produce the organic parts of their test walls, but were not able to produce new calcium carbonate. At the same time, there is no detectable change in the external pH during incubation with the Bafilomycin. These results are important in showing that proton pumping is directly responsible for the production of new calcium carbonate and we have added them to the revised version of our manuscript (lines 75-78, 101-107, 179-182, Table 1, Fig. S1, S2).

Bowman E. J., Siebers A. & Altendorf K. Bafilomycins: a class of inhibitors of membrane ATPases from microorganisms, animal cells, and plant cells. *Proc. Natl Acad. Sci.* 85, 7972-7976 (1988).

Also, It would be important to use isotope (or other) labels to identify the source of carbon that is integrated into the newly formed calcite as proof of the modeled mechanism.

Although we fully agree with the reviewer that such an experiment would help to pinpoint the route of the inorganic carbon, we have not included results from such an experiment in the revised version of our manuscript. Instead, we have elaborated on the isotopic consequences of the carbon uptake as we described in the original version of our

manuscript. The pathway followed by the inorganic carbon as a result of the observed high pH gradient inside/outside the foraminifer during calcification, matches very well with observed patterns in foraminiferal $\delta^{13}\text{C}$, for example, the observed decrease in $\delta^{13}\text{C}$ with increasing seawater $[\text{CO}_3^{2-}]$ (Spero et al., 1997). Moreover, the $\delta^{13}\text{C}$ of HCO_3^- is, at any pH, is always higher than that of the DIC (Zeebe and Wolf-Gladrow, 1999), which can thus not explain the $\delta^{13}\text{C}$ -signatures of foraminiferal calcite (<0). Instead, participation of $\text{CO}_2(\text{aq})$ is the only way to lower the $\delta^{13}\text{C}$ of the carbon in the calcium carbonate. For these reasons, we have included a discussion about influence of foraminiferal pH control on carbon isotopic signatures (see lines 217-230).

Spero, H. J., Bijma, J., Lea, D. W. & Bemis, B. E. Effect of seawater carbonate concentration on foraminiferal carbon and oxygen isotopes. *Nature* **390**, 497–500 (1997).
Zeebe, R. E. & Wolf-Gladrow, D. *CO₂ in seawater: Equilibrium, kinetics, isotopes*. Vol. 65 1-346 (Elsevier, 2001).

Finally, I think that the inferences made in this study should be put into a natural ecological context for these organisms. For example, questions that I have include: How do natural rates of water flow around the foraminifera alter the extracellular pH gradient? What levels of natural variation in DIC do foraminifera experience, and how would this study predict variation in calcification as a result?

The impact of the environment may be very different for the different species of foraminifera. Benthic species, as the reviewer suggests, may be subjected to considerably lower pH's than that of the bottom waters. Moreover, these pore water pH's may be variable in time (e.g. Stahl et al., 2006). Water movements may indeed alter the (3D-) shape and steepness of the gradients observed here, although this may be even stronger for planktonic species. Additionally, the effects of photosynthetic symbionts may also alter the pH gradients resulting from calcification. In summary, the gradients resulting from proton pumping may very likely differ between species and will depend on the environment. Therefore, it is very hard to spell out the *general* impact of ecology on the results presented here. We have therefore, included a short description of the processes that may alter the pH gradient caused by calcification (lines 261-268).

Stahl, H., Glud, A., Schroder, C. R., Klimant, I., Tengberg, A. & Glud, R. N. Time-resolved pH imaging in marine sediments with a luminescent planar optode. *Limnol. Oceanogr. Meth.* **4**, 336-345 (2006).

For a paper published in a high profile journal such as Nature, I think expectations for the completeness of an analysis (data-model-validation) is not too high.

According to the suggested improvements and our extension of the results and discussion, we think that the revised version of our manuscript provides a more complete and accurate description of the causes and consequences of foraminiferal proton pumping.

Reviewer #2 (Remarks to the Author):

Major Concerns:

1. I am unclear as to how many specimens the authors measured (line 76 mentions 5), but this should be clearly stated in the methods.

We have now added such basic information (line 290, 295), which has actually been increased due to one of the major concerns of reviewer #3 (incubations at various $p\text{CO}_2$'s).

2. I'm not sure what this paper has to do with ocean acidification. The forams do decrease in pH at the site of chamber formation and calcification, but this study didn't test for incubation at various pH's. I suggest taking out references to this, except in the discussion sections.

Also in answer to reviewer #3, we have now included results from incubations at a range of $p\text{CO}_2$'s. Another reason to keep in the relation of our results to ocean acidification, is that the relation between lowered external saturation state (due to OA) is relatively small compared to the acidification by the foraminifera themselves. This shows, contrary to what is generally assumed, that the relation between $p\text{CO}_2$ and shell production may not be as straightforward as often assumed (i.e. decreased omega results in less calcium carbonate production) (lines 93-100, 287-289, Table 1).

3. Did the authors test for the effect the pH indicator had on calcification? If this has been done previously, please cite.

We agree with the reviewer that this is necessary to test, and we now refer to De Nooijer et al. (2008) who showed that different species produced new chambers, pseudopodial activity and feeding in the presence of this fluorescent dye (lines 280-282).

de Nooijer, L. J., Toyofuku, T., Oguri, K., Nomaki, H. & Kitazato, H. Intracellular pH distribution in foraminifera determined by the fluorescent probe HPTS. *Limnol. Oceanogr. Meth.* 6, 610-618, doi:10.4319/lom.2008.6.610 (2008).

4. The implications of this study about calcification are modeled from inorganic carbonate chemistry measurements based on pH. I suggest the authors soften the language (Lines 183-185) to include the fact that these conclusions are from models, and not direct measurements of calcification.

We agree with the reviewer and have rephrased these sentences (lines 231-233).

Minor Concerns:

1. Please state what scale the indicator dye is measuring (NBS, seawater or total).

We have added the scale used for the indicator (seawater) (Table 1, Fig. 1, line81 in SI).

2. Please provide GPS points to where you collected your specimens, and in addition please provide information on culturing conditions before the observation (salinity, temperature, pH) if possible.

We have added the coordinates of our sampling (line 270) and have included the culturing conditions (line 277).

Reviewer #3 (Remarks to the Author):

Summary of key results

I'd like to start by congratulating the authors on conducting a set of simple but very elegant experiments using the pH sensitive fluorophore HPTS that visualize and quantify the proton efflux in the extracellular environment of foraminifers as they calcify new chambers. This is a novel set of results that makes a new and important contribution to understanding of biomineralization processes specifically in but not limited to foraminifers. It provides evidence for the likely use of 'massive' proton pumping to drive calcification, evidence to support a carbon concentration mechanism that could explain muted calcification responses to external environmental conditions, and also prima facie evidence for punctuated calcification with chamber addition in the studied specimens of *Ammonia* sp.

This is a highly original contribution and would be of general interest across the fields of climate change, biomineralization, cell biology, palaeoclimatology and palaeoceanography, among others.

Robustness and validity of Conclusions

Unfortunately the very real problem with the manuscript is the extent to which it overstates interpretations of findings as fact and asserts several unjustified conclusions about calcification responses to rising CO₂ levels. This is exemplified by the totally unsupported last sentence of the Abstract which states "These results suggest that increased CO₂ enhances calcification". Even if foraminifers did respond in this way, which most studies suggest they don't, the relatively small increase in surface ocean pCO₂ that would arise from a hypothetical increase in foraminifer calcification makes the following clause a gross overstatement of fact "thereby reducing the capacity of the ocean to take up atmosphere CO₂". To my knowledge virtually all studies of planktic and benthic perforate foraminifers show reduced (albeit slightly) calcification at higher pCO₂ and lower seawater pH and saturation state. Indeed this is acknowledged in the manuscript at line 187, where it is stated that many foraminifers exhibit moderate [negative] responses to experimentally induced ocean acidification. That the authors seem so keen to contradict and ignore this evidence in making their concluding statement is concerning.

We acknowledge the reviewer's concern and have toned down these statements. Even though our results may provide a mechanism for the moderate or even positive response of calcification related to enhanced [CO₂]/ lowered pH, it may still be that lowered saturation state may impair calcification and that these effects are (in some species) only partly mitigated by the positive effects of enhanced [DIC]. To reflect this complexity, we have rephrased the abstract and parts of the discussion (lines 26-28,231).

In relation to the above comments the first three sentences of the Abstract summarize some of the well-known feedback response of the ocean-atmosphere carbon cycle. This is not the topic of the paper, but rather seems to reflect an effort to give the manuscript greater 'global' significance. This is disingenuous and indeed unnecessary in my view. The advance in understanding biomineralization and perhaps also for designing future experiments that could better address more fundamental organism responses to changing ocean chemistry is an important step forward.

Following the reviewer's concern, we have deleted the first sentences of the abstract. Instead, we have focused more on the advancement in understanding foraminiferal biomineralization.

Continuing in the same vein; I would suggest rewording of concluding paragraph beginning line 182. "Our results show that calcification is not directly coupled to the abundance of carbonate ions and saturation state of seawater." There then appears to be a limited basis for making the following statement, as there is no evidence supplied to suggest that calcification relies on the total inorganic carbon concentration, as much as this might or might not be true.

Possible additional experiments that address foraminifer responses to seawater DIC and saturation state?

We agree and have rephrased the suggested section of the discussion (line 231). As we answered to reviewer #1, we have added results on decreases in external pH during calcification under different $p\text{CO}_2$'s (lines 93-100, Table 1).

The manuscript seems driven to generate a headline that states foraminifer calcification is not negatively affected by rising atmosphere $p\text{CO}_2$ and decreasing calcite saturation state, however there is little or no evidence generated here to justify these assertions nor to evaluate response to changes in foraminifer proton efflux responses to rising $p\text{CO}_2$ and decreasing saturation state. Based on the description of the experimental method it should be simple enough to undertake a small set of experiments to test how the visualized proton efflux (and calcification) changes with the DIC and pH of the seawater medium. This would make for an more interesting study and would address some of the more fundamental responses that foraminifers exhibit to rising $p\text{CO}_2$ and decreasing saturation state.

There are indeed very mixed responses between species to decreased saturation state/ lowered pH/ increased CO_2 . Our results lead logically to the question whether all species make use of this proton pumping (and if, whether they decrease the external pH to the same extent). If that is not the case, it may well explain the mixed responses in terms of growth and calcification under OA scenarios. Related to this and previous comments of the reviewer, we have adjusted the text of the discussion to reflect this complexity (lines 241-251).

Treatment of data and uncertainties, and interpretation of results

I wonder whether there is the possibility of a chicken-and-egg situation with the results. Specifically, could an equivalent proton flux be generated through calcification (driven by a mechanism other than proton pumping) combined with enhanced respired CO₂ production during chamber formation. How would the observed [H⁺] distribution around the foraminifer differ from this alternate scenario? Is a second observable in the seawater carbonate system needed to distinguish between these possibilities, or can the proton pump mechanism only explain the data?

Thanks to this comment and the first concern by reviewer #1, we have added results of foraminifera incubated with an inhibitor targeting V-type H⁺ATPases (Bafilomycin A1). After addition of this inhibitor at the onset of calcification, external proton fluxes were not clear and almost no new calcium carbonate was formed (lines 75-78, 101-107, 171-176, Table 1, Fig. S1, S2). This shows that the reduction in external pH is not a consequence of e.g. CO₂ release, but that it is directly responsible for calcification. These results strengthen our theory that calcification in foraminifera is achieved by exchanging H⁺ for Ca²⁺.

What are the uncertainties of the proton flux and total flux QH estimates?

Understanding the efficiency of converting CO₂ and HCO₃⁻ within the site of calcification (SOC) and thus the efficiency with which inorganic carbon is used for calcium carbonate production would be useful in the context of the proposed carbon concentration mechanism. How does the QH compare to the amount (in moles) of calcite produced during chamber addition? Specifically, does it account for the amount of calcite precipitated.

The uncertainty in the proton flux is mainly caused by the uncertainty in the determined pH. Additionally, spatial variability in reduced pH, the estimated volume over which the pH gradient is modelled and assumptions in the modeling, increase the uncertainty in the calculated flux. Therefore, and as we state in the revised version of our manuscript, the calculated fluxes are not very precise. As the reviewer suggests, the flux is particularly interesting in comparison to the total inward flux of carbon (and Ca²⁺). In the manuscript we refer to estimates for the total flux of carbon and calcium in similar-sized specimens of the same genus (De Nooijer et al., 2009b). (line 123) Since these estimates are based on the volume of calcite precipitated, the uncertainty in these fluxes result from the accuracy of the wall depth, chamber shape and pore density. Meantime, the specimen continues a hardening of chamber wall even after chamber formation. Since these uncertainties add up to a relatively large uncertainty in the carbon- and calcium flux, it becomes challenging to directly relate the fluxes of all cations during calcification for future work. We therefore have kept the notion that the fluxes appear to be of the comparative order of magnitude and refrain from suggesting that the fluxes are more similar than we currently know.

de Nooijer, L. J., Langer, G., Nehrke, G. & Bijma, J. Physiological controls on seawater uptake and calcification in the benthic foraminifer *Ammonia tepida*. *Biogeosciences* 6, 2669-2675, doi:10.5194/bg-6-2669-2009 (2009).

The model being proposed is that the proton efflux is used to raise the fCO₂ in the surrounding microenvironment to a level where it diffuses into the cell and SOC. While feasible, there is no direct evidence supporting this proposal. It would require external fCO₂ being > internal fCO₂ of the pseudopodial envelope in your model. This might be the case

but the internal cellular $f\text{CO}_2$ would likely be quite high and so would create a partial barrier (slow point) that would not be particularly conducive to driving a large CO_2 flux from the high $f\text{CO}_2$ seawater to the low $f\text{CO}_2$ SOC. How accurate are the pH estimates obtained from the HPTS fluorophore calibration?

The reviewer is correct that we indeed infer an influx from inorganic carbon (CO_2 , effectively) into the SOC. The nature and thickness of the pseudopodial network will indeed determine the diffusion rate during calcification. To answer the last question of this reviewer, reproducibility of the ratiometric pH determination is better than ± 0.15 units (see also De Nooijer et al., 2008). This means that the internal (De Nooijer et al., 2009a) versus external (this study) pH is well above 2 units, creating a very steep gradient in $p\text{CO}_2$ over a very short distance. We therefore infer that inward CO_2 diffusion (see also Bentov and Erez, 2006 for a similar intracellular process in foraminifera) likely provides the carbon necessary for calcification. At the same time, the transformation of CO_2 to carbonate thereby provide the protons necessary to continue the outward H^+ flux. We are aware, however, that strictly speaking this is an assumption and have therefore made sure that this is reflected in the text (line 196-216).

de Nooijer, L. J., Toyofuku, T., Oguri, K., Nomaki, H. & Kitazato, H. Intracellular pH distribution in foraminifera determined by the fluorescent probe HPTS. *Limnol. Oceanogr. Meth.* **6**, 610-618, doi:10.4319/lom.2008.6.610 (2008).

de Nooijer, L. J., Toyofuku, T. & Kitazato, H. Foraminifera promote calcification by elevating their intracellular pH. *Proc Natl Acad Sci U S A* **106**, 15374-15378, doi:10.1073/pnas.0904306106 (2009).

Bentov, S. & Erez, J. Impact of biomineralization processes on the Mg content of foraminiferal shells: A biological perspective. *Geochem. Geophys. Geosy.* **7**, Q01P08, doi:10.1029/2005GC001015 (2006).

In relation to the model I note that there is also no evidence to substantiate the use of Ca^{2+} , which otherwise implies a selective process for Ca uptake from external seawater in Figure 3 (as opposed seawater itself as a source of ions).

We only partly agree with the reviewer on this point. Nehrke et al. (2013) showed that there is an inward Ca^{2+} influx in this foraminiferal species, which moreover (De Nooijer et al., 2009b), roughly matches the estimated proton flux reported in our manuscript. We do not know the exact mechanism by which the Ca-ions are transported into the SOC, but the large pH gradient suggests that Ca^{2+} is transported across the membranes separating the SOC from the surrounding medium. Based on these published results, we suggest to leave in this flux in our conceptual model (line 194, Fig. 3).

Nehrke, G. et al. A new model for biomineralization and trace-element signatures of Foraminifera tests. *Biogeosciences* **10**, 6759-6767, doi:10.5194/bg-10-6759-2013 (2013).

Overall the manuscript is well written and in a style that is easily understandable.

We appreciate the detailed comments and discussion.

Additional minor comments

Chamber is typed incorrectly in Table 1

We corrected this typo.

Line 132 .. should read .. Inside the site of calcification, elevated pH

K_a is noted to be the second type of the modified Bessel function but does not appear as such in (E.2).

We have found that there were two problems in our original manuscript.

One is that we used an italic faced K in E.2 but a Roman K was used in the explanation.

The other one is that $K_{1/2}$ was used in E.2 and K_a was used in the explanation, where we meant that the case $a=1/2$ was assumed in E.2. In the revised manuscript, we have modified the K to the italic face in the explanation and specified that $a=1/2$ is assumed.

(lines 49-52 in supplementary information)

REVIEWERS' COMMENTS:

Reviewer #1 (Remarks to the Author):

I have read through the author's response to reviewers and find all of the responses to be adequate. I was particularly pleased to see the authors perform additional experiments with Bafilomycin, and find those results to be convincing evidence in support of the proposed mechanistic hypotheses.

At this point, I find the manuscript ready for publication.

Reviewer #2 (Remarks to the Author):

I would like to congratulate the authors on a well written and important piece of work that is produced. The authors have sufficiently answered all my previous comments and concerns, and I support this manuscript for submission to Nature Communications.

Reviewer #3 (Remarks to the Author):

I commend the authors on their revised manuscript; in particular the new experimental results shown using the V-type proton pump blocker. I am satisfied with the vast bulk of the changes made to the manuscript by the authors in response to the reviewer comments. The links drawn between the stoichiometry of the total H⁺ efflux and Ca²⁺ required for calcification are interesting but do not provide direct evidence for Ca²⁺ manipulation by the foraminifers ... this is a hypothesis that remains to be tested.

I make the following detailed minor corrections and suggestions:

Line 26-28 Suggest qualifying the following sentence with potentially

These results also suggest that increased CO₂ may not reduce calcification, thereby potentially maintaining the current global marine carbonate production.

Lines 44-47 I found the latter part of the following sentence confusing

The key to understanding foraminiferal calcification centers on the relation between carbon speciation in seawater and which species, CO₂ and the bicarbonate and preferential uptake of these chemical species (CO₂, bicarbonate and/or carbonate ions)

Line 52 Suggest adding may to the following sentence

Limited availability of carbonate ions in seawater dictates that foraminifera may require unrealistic volumes of seawater to produce new calcium carbonate

Line 89 suggest 'closest to each organism and in particular'

Line 118 suggest using 'a proportion' instead of 'part' in this sentence ...

Line 130-131 suggest replacing "With a new chamber in the shape of half a sphere" with "For a hemispherical chamber with a diameter of 20-50 microns ... etc

Line 134-136 Suggest the following reword Figure 1. Representative images showing the time-resolved decrease in pH of seawater surrounding a calcifying specimen of *Ammonia aomoriensis*. Over a period of 320 mins. The pH values are imaged using dissolved HPTS and reported on the seawater scale.

Line 156..... by theoretical fitting (of what????)

Line 226 suggest ... is similar to that proposed by previous studies

Line 255 use shown instead of show

Line 263 the timescale of hours

Finally, the figure captions tend to be rather poorly written, I suggest some attention to be given to these prior to publication

Response to Referees' letter

We sincerely thank the all reviewers for constructive comments, which were of great help in improving the manuscript.

We have revised the figure captions and incorporated all other suggestions from you and reviewer #3. This includes an explanation of the error bars in Figure 2 and a rephrasing of the suggestion that Ca^{2+} - and proton-flux stoichiometry is evidence for their direct coupling during biomineralization (lines 146-147).

Reviewer #3 (Remarks to the Author):

I commend the authors on their revised manuscript; in particular the new experimental results shown using the V-type proton pump blocker. I am satisfied with the vast bulk of the changes made to the manuscript by the authors in response to the reviewer comments. The links drawn between the stoichiometry of the total H^+ efflux and Ca^{2+} required for calcification are interesting but do not provide direct evidence for Ca^{2+} manipulation by the foraminifers ... this is a hypothesis that remains to be tested.

I make the following detailed minor corrections and suggestions:

Line 26-28 Suggest qualifying the following sentence with potentially
These results also suggest that increased CO_2 may not reduce calcification, thereby potentially maintaining the current global marine carbonate production.

Lines 44-47 I found the latter part of the following sentence confusing
The key to understanding foraminiferal calcification centers on the relation between carbon speciation in seawater and which species, CO_2 and the bicarbonate and preferential uptake of these chemical species (CO_2 ,

bicarbonate and/or carbonate ions)

Line 52 Suggest adding may to the following sentence

Limited availability of carbonate ions in seawater dictates that foraminifera may require unrealistic volumes of seawater to produce new calcium carbonate

Line 89 suggest 'closest to each organism and in particular'

Line 118 suggest using 'a proportion' instead of 'part' in this sentence ...

Line 130-131 suggest replacing "With a new chamber in the shape of half a sphere" with "For a hemispherical chamber with a diameter of 20-50 microns ... etc

Line 134-136 Suggest the following reword Figure 1. Representative images showing the time-resolved decrease in pH of seawater surrounding a calcifying specimen of *Ammonia aomoriensis*. Over a period of 320 mins. The pH values are imaged using dissolved HPTS and reported on the seawater scale.

Line 156..... by theoretical fitting (of what???)

Line 226 suggest ... is similar to that proposed by previous studies

Line 255 use shown instead of show

Line 263 the timescale of hours

Finally, the figure captions tend to be rather poorly written, I suggest some attention to be given to these prior to publication